# Tyrosinase Inhibitory Peptides from Enzyme Hydrolyzed Royal Jelly: Production, Separation, Identification and Docking Analysis

**DOI:** 10.3390/foods12112240

**Published:** 2023-06-01

**Authors:** Zhen Ge, Jun-Cai Liu, Jian-An Sun, Xiang-Zhao Mao

**Affiliations:** 1Qingdao Key Laboratory of Food Biotechnology, College of Food Science and Engineering, Ocean University of China, Qingdao 266404, China; gezhen2024@163.com (Z.G.); liujuncai07@163.com (J.-C.L.); xzhmao@ouc.edu.cn (X.-Z.M.); 2Key Laboratory of Biological Processing of Aquatic Products, China National Light Industry, Qingdao 266404, China; 3Laboratory for Marine Drugs and Bioproducts of Qingdao National Laboratory for Marine Science and Technology, Qingdao 266237, China

**Keywords:** royal jelly peptides, Alzheimer’s disease, tyrosinase, LC-MS/MS, molecular docking

## Abstract

Tyrosinase is inextricably related to the development of Alzheimer’s disease. The effects of natural tyrosinase inhibitors on human health have attracted widespread attention. This study aimed to isolate and analyze the tyrosinase (TYR) inhibitory peptides in the enzymatic digestion products of royal jelly. We first analyzed optimal process conditions for the enzymatic digestion of royal jelly by single-factor and orthogonal experiments and then used gel filtration chromatography to obtain five fractions (D1~D5) with molecular weights ranging from 600 to 1100 Da. LC-MS/MS was applied to identify the fractions with the highest activity, and the obtained peptides were screened and molecularly docked using AutoDock Vina. The results showed that the optimal enzymatic conditions for tyrosinase inhibition rate were acid protease, enzyme addition 10,000 U/g, initial pH 4, feed-to-liquid ratio 1:4, enzymatic temperature 55 °C, and enzymatic time 4 h. The D4 fraction had the most significant TYR inhibitory activity. The IC_50_ values of the three new peptides with the strongest TYR inhibitory activity, TIPPPT, IIPFIF, and ILFTLL, were obtained as 7.59 mg/mL, 6.16 mg/mL, and 9.25 mg/mL, respectively. The molecular docking results showed that aromatic and hydrophobic amino acids were more favorable to occupy the catalytic center of TYR. In conclusion, the new peptide extracted from royal jelly has the potential to be used as a natural TYR inhibitory peptide in food products with health-promoting properties.

## 1. Introduction

Worldwide, dementia is the second most common neurological ailment, following various types of headaches, with Alzheimer’s disease (AD) accounting for 70–80% of dementia cases [1,2]. AD is an irreversible neurological illness that causes cognitive and memory loss, mental and behavioral disorders, and language impairments, all of which can severely affect a person’s social and professional life [3,4]. Studies have shown that Aβ deposition, neurofibrillary tangles, oxidative stress, cholinergic deficiency, and neuroinflammation are factors in the formation of AD [5,6,7]. To slow the progression of AD, clinical studies have used enzymes (including β-secretase and γ-secretase, which are involved in amyloid precursor protein catabolism and acetylcholinesterase, which is involved in acetylcholine catabolism) as targets to attempt drug design due to alterations in enzyme activity, which usually have definite effects [8]. Unfortunately, these medications have numerous side effects [9].

Tyrosinase (TYR), a key rate-limiting copper-containing enzyme that controls the amount of melanin produced during melanogenesis, may catalyze two reactions simultaneously by first hydroxylating L-tyrosine to 3,4-Dihydroxy-Lphenylalanine (L-DOPA) and then oxidizing L-DOPA to make dopaquinone [10,11]. TYR is not only intrinsically linked to the development of age spots, photodamage, and skin pigmentation [12], but also to the development of AD. According to studies, elevated amounts of L-DOPA, an intermediate substance generated through the action of TYR during the manufacture of melanin, can cause neurotoxicity, inflammatory reactions [13,14], as well as a tangential rise in tau protein phosphorylation, in living organisms [15]. These are all risk factors that contribute to the progression of AD. In addition, TYR activity could be higher in substantia nigra and involved in neuromelanin synthesis [9], but its exact role in the brain is not known. Targeted inhibition of TYR may help to block the AD process; therefore, research into TYR inhibitors is critical in the food and pharmaceutical industries.

Currently, arbutin, hydroquinone, and kojic acid have strong inhibitory effects on TYR activity; however, these inhibitors are often limited due to their instability, low permeability, and high toxicity to cells [16,17]. Bioactive peptides, as a new type of therapeutic drug, have attracted a lot of attention in recent years. Bioactive peptides are functional peptide fragments produced during the hydrolysis, cleavage, or maturation of proteins, and their relative molecular mass is usually less than 6000 Da [18,19]. Tyrosinase inhibitory peptides (TIPs) generated from food are mostly utilized in food, pharmaceutical and skin care applications [20,21,22], particularly for whitening and photodamage restoration, and are appreciated for their excellent biosafety and ease of absorption. Food protein-derived TIPs have strong biocompatibility and are prevalent in mammalian milk [23], agricultural products [24], and aquatic products [25], primarily from terrestrial and aquatic sources. TIPs are currently prepared using enzymatic digestion, chemical hydrolysis, microbial fermentation, and chemical synthesis, with enzymatic digestion and solid-phase synthesis being the two most commonly used techniques [6]. Separation and purification methods primarily include ultrafiltration [26,27], gel filtration chromatography [28,29], and reversed-phase high-performance liquid chromatography [28,30]. In addition, researchers have predicted that peptides may occupy the position of the substrate by chelating with copper ions or binding to surrounding amino acid residues, ultimately inhibiting the catalytic activity of TYR [31].

Royal jelly (RJ) is a secretion from the hypopharynx and mandibular glands of young worker bees. Fresh royal jelly mainly consists of water (60–70%), protein (9–18%), carbs (7–18%), lipids (3–8%), minerals (1–3%), vitamins, and a few unidentified substances [32], and its protein content accounts for 30–50% of the dry matter, which has high nutritional value. However, the sensory characteristics of RJ, such as pungent and sour taste, sticky texture, and low solubility, restrict its processing and utilization and economic value in the market. RJ peptides have been reported to have multiple active functions [33,34]. Acid protease, papain, alkaline protease, and other proteases are often employed for the enzymatic digestion of royal jelly. They are all endopeptidases that can cleave peptide bonds generated by a wide range of amino acids. Acid protease, for illustration, can cleave the peptide link between aromatic or hydrophobic amino acid residues on both ends [35]. Gu found that enzyme-treated royal jelly (ERJ) has some antioxidant capacity due to the mechanism of reduction of intracellular reactive oxygen species (ROS) and nitric oxide (NO) production in LPS-treated macrophages. In addition, ERJ significantly increased the activity of the antioxidant enzyme superoxide dismutase (SOD) in a dose-dependent relationship with antioxidant glutathione (GSH) levels [36]. Zhang used gastrotrypsin to hydrolyze the main protein of RJ and purified and isolated the hydrolysis products, which were obtained with a molecular weight of <1 ku with strong angiotensin-converting enzyme (ACE) inhibitory activity (IC_50_ = 0.33 mg/mL) [37]. Zhang found that royal jelly peptides (RJPs) have neuroprotective functions and that purified RJPs inhibit β-amyloid 40 and β-amyloid 42 production in N2a/APP695 cells by downregulating β-secretase (BACE1), the regulation of which may be associated with histone acetylation modifications [38]. However, there is a lack of research on the TYR inhibition mechanism of RJPs.

Molecular docking is a crucial computer-aided tool in structural molecular biology drug design, allowing the binding of small molecule ligands to large molecule receptors to be simulated and visualized [39]. Molecular docking can be used to study the affinity and mechanism of the action of molecules with receptors, can identify the most active ligands, and can propose structural hypotheses for ligand inhibition of related receptors [40]. In recent years, molecular docking due to its high efficiency and low cost has been commonly utilized for screening and discovering natural small molecule active chemicals.

This study combined traditional experiments with computer-aided techniques to obtain potent TYR inhibitory peptides from the enzymatic digestion products of RJ. The enzymatic digestion of royal jelly (ERJ) was optimized with a single-factor experiment and response surface, and the purified and separated fraction with the greatest TYR inhibitory activity was obtained by gel filtration chromatography. LC-MS/MS and molecular docking were used to identify, evaluate, and screen highly active TYR inhibitory peptides. The present work is expected to provide a basis for the development of novel, safe, and effective TYR inhibitory peptides.

## 2. Materials and Methods

### 2.1. Materials and Reagents

The fresh royal jelly (freeze-dried powder with 47.5% protein content) was provided by Shandong Fengcai Health Industry Co., Ltd. (Weifang, Shandong, China). Acid protease (100,000 U/g) was purchased from Pangbo Biological Engineering Co., Ltd. (Nanning, China). Sephadex G-15, glutathione, bacitracin, insulin, and cytochrome C were purchased from Solarbio Science & Technology Co., Ltd. (Beijing, China). TYR inhibitory peptides were synthesized by Sangon Biotech Co., Ltd. (Shanghai, China). TYR was purchased from Shanghai Yuanye Bio-Technology Co., Ltd. (Shanghai, China). HPLC grade reagents were from Merck Drugs & Biotechnology, and other chemical reagents were analytical grades.

### 2.2. Preparation of ERJ

Based on the single-factor test, the three levels of the significant parameters and the interaction effects between various factors, including enzymatic digestion temperature, solid-liquid ratio, and enzyme dose, which influence the value of inhibitory activity against TYR significantly, were analyzed and optimized by Box–Behnken methodology of Design Expert 10 software (Appendix A). The optimal model conditions were obtained as follows: acid protease, material-liquid ratio 1:4, temperature 55 °C, enzyme addition 10,000 U/g, initial pH 4.0, and enzymatic digestion time 4 h, and the predicted TYR inhibition rate was 88.88%, the response surface optimization is shown in Appendix A. The temperature was controlled with a THZ-82 thermostatic water bath shaker (Shanghai Zhi Chu Instruments Co., Ltd., Shanghai, China), and the gyration frequency was kept constant at 200 rpm/min throughout the enzymatic digestion process. The response surface optimization model was validated, and the actual measured TYR inhibition rate was 90.80%, which was close to the predicted value, indicating that the regression equation can reflect the influence of each factor on the process and has practical value.

The RJ was enzymatically digested under the best experimental conditions obtained by response surface optimization. After enzymatic digestion, the ERJ was boiled at 100 °C for 12 min to deactivate the enzyme.

### 2.3. Determination of Inhibitory Activity of TYR

Determination and calculation methods of TYR inhibitory activity were referred to the detection method of Ji and modified slightly [41]. Briefly, the TYR was dissolved to prepare solutions using 0.05 M phosphate buffer saline (pH 6.8). Then, the mixed 40 μL of ERJ, 2 U/mL 100 μL of TYR, and 1.0 mmol/L 100 μL of L-DOPA were added in a 96-well plate, and the absorbance at 475 nm was measured immediately after 10 s and recorded as A1. The reaction system was incubated for 190 s at 25 °C and measured again and recorded as A2. We repeated the above steps by replacing the ERJ with buffer and measuring the absorbance value after 10 s, recorded as A3, and the absorbance value after 190 s as A4. The inhibition rate of ERJ on TYR was calculated according to the following equation:(1)$$\mathrm{Inhibition}\;\mathrm{percentage}\;(\%)=(1-\frac{\Delta A}{\Delta B})\;\times \;100\%$$
where, ΔA = A2 − A1, ΔB = A4 − A3.

### 2.4. Molecular Weight Determination

The molecular weight distribution of ERJ was determined by BIOTECH-30BS High Performance Liquid Chromatography (HPLC) (Agilent Technologies, Inc., Santa Clara, CA, USA) [42]. Chromatographic columns: TSKgel G2000SWXL (300 × 7.8 mm); mobile phase: acetonitrile: water: trifluoroacetic acid = 45:55:0.1; detection ultraviolet wavelength: 220 nm; flow rate: 0.5 mL/min; column temperature: 30 °C. The logarithmic connection between retention durations and relative molecular weights of glutathione, bacitracin, insulin, and cytochrome C was used to generate the standard curves. The molecular weight distribution of ERJ can be calculated using the standard curve and the peak time.

### 2.5. Purification of Inhibitory Peptide

Sephadex G-15 was used as the gel filtration packing, which was swollen after sufficient washing and injected into the 16 mm × 70 cm column (Solarbio Science & Technology Co., Ltd., Beijing, China), with continuous injection along the wall to avoid delamination and air bubbles. The 200 mg/mL sample was filtered by 0.45 mm microfiltration membrane, and the 1 mL of filtered sample was injected into the chromatographic column. Finally, the different components were collected according to the chromatographic peaks, freeze-dried separately (Xinzhi Biotechnology Co., Ltd., Ningbo, China), and the TYR inhibition activity of each component was determined.

### 2.6. Identification of the Highest Active Component by LC-MS/MS

#### 2.6.1. Extraction of Polypeptides

The most active fraction was redissolved with 0.1% trifluoroacetic acid (TFA) and centrifuged for 15 min at 12,000× *g* (5804R frozen centrifuge: Eppendorf Co., Ltd., Hamburg, Germany) in a 30 KDa ultrafiltration tube. The filtrate was transferred to a new 1.5 mL EP tube and desalted using C18 Stage Tip and dried in vacuo. Finally, the dried peptides were dissolved in 0.1% formic acid aqueous solution for LC-MS/MS identification.

#### 2.6.2. LC-MS/MS Analysis of Component

The method refers to Zhao’s [43], with appropriate modifications. Thermo Scientific’s Easy nLC 1200 chromatography system with a nanoliter flow rate was used to extract the 1.2 g of peptide from the sample. Buffer: formic acid of 0.1% in an aqueous solution comprised solution A, while solution B was a combination of 0.1% formic acid, acetonitrile, and water, with acetonitrile making up 80% of the mixture. Then, 100% of solution A was used to balance the column. Samples were injected into Trap Colum (100 μm × 20 mm, 5 μm, C18, Dr. Maisch GmbH, Ammerbuch, Germany) and then separated by a chromatographic column (75 μm × 150 mm, 3 μm, C18, Dr. Maisch GmbH, Ammerbuch, Germany) at a flow rate of 300 nL/min. The gradient of liquid phase separation was as follows: from 0 to 2 min, the B liquid linear gradient ranged from 2% to 5%; from 2 min to 44 min, the B liquid linear gradient ranged from 5% to 28%; from 44 min to 51 min, the B liquid linear gradient ranged from 28% to 40%; from 51 min to 53 min, the B liquid linear gradient ranged from 40% to 100%; and from 53 min to 60 min, the B liquid was maintained at 100%. Peptide isolation was followed by DDA (data-dependent acquisition) mass spectrometry analysis using the Thermo Scientific Q-Exactive Plus mass spectrometry equipment. The analysis lasted 60 min, the detection mode was positive ions, the mother ion scanning range was 350–1800 *m*/*z*, the main mass spectrometry resolution was 60,000 @*m*/*z* 200, the AGC target was 3e6, and the first level Maximum IT was 50 ms. The following procedure was used to obtain peptide secondary mass spectrometry: triggered the acquisition of a secondary mass spectrometry (MS2 scan) of the 20 highest intensity parent ions after each full scan, secondary mass spectrometry resolution: 15,000 @*m*/*z* 200, AGC target: 1e5, secondary Maximum IT: 50 ms, MS2 Activation Type: HCD, Isolation window: 1.6 *m*/*z*, Normalized Collision Energy: 28.

#### 2.6.3. Database Search of Peptides

pFind was chosen as the mass spectrometry database search software; the following protein database was used: UniProt-Apis mellifera (Honeybee) [7460]-40765-20220524. The pFind library search software analysis parameters were set as shown in Table 1.

### 2.7. Peptide Screening and Molecular Docking

In this study, the peptide–protein interaction was assessed via molecular docking using the free software AutoDock Vina 1.1.2, and the results were represented as Vina score (kcal/mol), which reflects the binding affinity of TYR to peptide, the stronger the binding affinity, the lower the Vina score. The macromolecular receptors preparation was referred to by Li [44]. The TYR X-ray crystal structure (PDB ID: 2Y9X) can be downloaded from the Protein Data Bank (https://www.rcsb.org/, accessed on 6 July 2011) database. We removed the water molecules, B chain, and other ligands from 2Y9X using the PyMOL program and saved it in PDBQT format. The polypeptide’s three-dimensional structure was created with ChemDraw 2019 and Chem3D Pro 14.0, saved as MOl2 format after MM2 field optimization, and then converted to PDBQT format by AutoDockTools program and saved after hydrogenation, torsional center, and torsional key identification.

The docking process used the AutoDock Vina, with reference to the position of the TYR in 2Y9X, setting the docking center coordinates to (−2.927, 22.019, −32.487) (x, y, z), the docking box size to 110 × 100 × 110, the docking grid spacing to 0.43A, and other parameters to the default values. The polypeptide ligands were individually docked to the PDBQT format TYR crystal structure. The interaction of ligands and receptors was demonstrated using the PyMOL and AutoDockTools software.

### 2.8. Peptide Synthesis

All peptides were synthesized by Fmoc solid-phase synthesis method in Sangon Biotech Co., Ltd. (Shanghai, China).

### 2.9. Statistical Analysis

All experiments were performed in three replicates and values were expressed as mean ± standard deviation. Origin 2018 software was used for plotting and SPSS 27.0 (IBM Corp, Armonk, NY, http://ms.ouc.edu.cn/, accessed on 6 July 2011) software was used for experimental statistical analysis (ANOVA).

## 3. Results and Discussion

### 3.1. Separation and Purification of ERJ

Using proteases to hydrolyze endogenous proteins is an effective and low-cost method for producing biologically active peptides with specified functions. In this study, proteins from RJ were used to extract active peptides having TYR inhibitory activity.

The crude enzymatic product was obtained by enzymatic digestion of RJ under the optimized conditions, and its TYR inhibition rate was 90.80%, which was close to the predicted value. Kim and Park demonstrated that the two major royal jelly proteins, MRJP-2 and MRJP-4, have antibacterial properties and can be used to sterilize organisms by attaching to fungus, yeast, and bacteria and rupturing their cell walls [45,46]. Guo isolated and purified 29 antioxidant peptides from RJ and verified their antioxidant activity. The results showed that 12 small peptides with two to four residues had strong activity, among which three dipeptides, Lys-Tyr, Arg-Tyr, and Tyr-Tyr, showed strong free radical scavenging ability due to the hydrogen atoms of their phenolic hydroxyl groups [47]. These findings show that RJ is a great source of protein for making bioactive peptides and that enzymatic digestion is a good way to make bioactive peptides. In general, the composition of crude enzymatic digestion products is complex, and the purification and identification of peptide sequences with TYR inhibitory activity from crude enzymatic digestion products is a critical research topic. The complexity of the enzymatic digestion product is decreased through the use of peptide isolation or enrichment procedures, which makes it easier to acquire the target peptide.

Gel filtration chromatography was utilized in this study to isolate and purify the component ERJ. We first identified the type of packing used for gel filtration chromatography by measuring the range of molecular weight distribution of ERJ. Figure 1 depicts the molecular weight peaks of the standards versus retention time, as well as the ERJ molecular weight distribution. The standard curve equation is fitted as y = −0.4019*x + 6.5125, R^2^ = 0.9797. The results demonstrated that the molecular weight distribution in the ERJ in the range of 39.329~867.034 Da accounted for 82.62% of the whole sample, and the molecular weight distribution was primarily concentrated below 1000 Da. Thus, Sephadex G-15 was chosen as the chromatographic column packing to achieve the separation of small molecule peptides.

The enzymatic digest under the optimal process conditions was prepared as the sample for loading, and the loading concentration and volume were: 200 mg/mL, 1 mL. The five peaks were collected and named D1 to D5 (Figure 2). The separate elution peaks were collected, lyophilized, and compounded into a solution with a peptide concentration of 21 mg/mL, and the TYR inhibition rates of the different fractions varied widely. As shown in Table 2, the TYR inhibition rate of ERJ (D0) was 7.22%, while the inhibition rate of fraction D4 reached 82.89% which was significantly higher than the other fractions (*p* < 0.01). Therefore, fraction D4 was selected for further bioactive peptide identification.

### 3.2. Identification and Virtual Screening of Potential Bioactive Peptides

LC-MS/MS was used to identify the amino acid sequences of the D4 fraction with the greatest activity, and the base peaks of the samples are displayed in Figure 3. Using PSM FDR ≤ 0.01 and Protein FDR ≤ 0.01 as screening criteria for peptide and protein identification, mass spectrometry data were searched for D4 fractions, then 41 peptide sequences and 47 protein sequences were obtained, respectively, with most peptides carrying one or two charges and molecular weights in the range of 603~1058 Da. Interestingly, most of these peptides include between five and eight and no more than 10 amino acids. In addition, 85% of the peptides contained hydrophobic amino acids at the c-terminus. Acid proteases extensively cleave the peptide bonds between hydrophobic or aromatic amino acids, which explains the larger proportion of the obtained peptides containing hydrophobic amino acids at the ends [35]. We utilized Peptide Ranker to rank the predicted scores of peptide bioactivity and screened 10 peptides with scores greater than five-tenths to determine the possibility of these bioactive peptides.

The AutoDock Vina program was used to dock peptides with the TYR to quickly identify TYR inhibitory peptides with greater activity from these 41 peptides. The results showed that the binding energies of 41 peptides to TYR were between −5.2 kcal/moL and −7.5 kcal/moL, of which 11 peptides had binding energies less than −6.9 kcal/moL and eight peptides had binding energies less than −7.0 kcal/moL. A lower binding energy indicates that the peptide is more stably bound to the receptor protein. However, not all peptides with a Peptide Ranker score greater than five-tenths have a low binding energy to TYR. The Peptide Ranker score only shows the potential for physiological activity. Seven peptides with low binding energies (≤−6.80 kcal/moL), as well as Peptide Ranker scores ≥ 0.5, were ultimately chosen for synthesis and activity validation. It was verified that all the above seven peptides had TYR inhibitory effects, among which three peptides, TIPPPT, IIPFIF, and ILFTLL, showed the strongest inhibitory activity, indicating that AutoDock Vina could significantly improve the screening efficiency of the peptides. Table 3 lists the molecular weight and the IC_50_ values for TYR inhibition of three peptides. The maximum activity was found with the peptide IIPFIF, followed by TIPPPT, and the lowest activity was found with the peptide ILFTLL. The obtained peptide had similar TYR inhibitory action to VLT (IC_50_ = 5.0 mg/mL) [48].

### 3.3. Peptide and TYR Interaction Analysis

To further understand the mechanism of peptide inhibition on TYR, we used molecular docking to explore the interaction between TIPPPT, IIPFIF, ILFTLL, and TYR, and the docking findings are presented in Figure 4.

The comparative results showed that the three peptides bind effectively to the TYR peripheral residues mainly through hydrogen bonds, van der Waals forces, hydrophobic interactions, and π-σ bonds. Ismaya analyzed the crystal structure of the mushroom tyrosinase (abTYR) isolated from *Agaricus bisporus*. The crystal structure of abTYR is an H2L2 tetramer consisting of two H subunits and two L subunits [49]. The L subunit contains 150 subunits and is the product of an unrelated gene with an unknown function; whereas the H subunit is thought to be derived from the TYR ppo3 gene because it includes 391 amino acid residues that are identical to residues 2–392 of ppo3 [50]. Wu found that TYR high-activity inhibitors usually contain groups such as thiosemicarbazide, amide group, triazole, thiophene, thiadiazole, and kadiridine, and low-activity inhibitors usually include groups such as piperidine, benzoheterocycle, carboxyl group on aromatic ring, morpholine, oxygen-containing aliphatic ring, and sulfur-containing five-membered aliphatic ring [51].

The active binding site of abTYR is located on the H subunit and consists of two copper atoms (CuA and CuB) and six histidines (His). CuA binds to the His61, His85, and His94 ligands, while CuB binds to the His259, His263, and His296 ligands. The result of these histidine residues being ligated with copper ions is that their rotational freedom is restricted. To maintain the integrity of the binding site, Phe90, which is situated between His94, His259, and His296, and Phe292, which is situated between His61, His263, and His296, further constrain the histidine side chain conformation [39]. Therefore, His61, His85, His94, His259, His263, His296, Phe90, and Phe292 are crucial amino acid residues that impact TYR activity. As shown in Figure 4, three peptides interact with at least one important amino acid. Peptide TIPPPT forms five conventional hydrogen bonds with Asn260, Asn81, Tyr65, Tyr78, and Ala23 of TYR, van der Waals forces with His263, His259, Ala80, Arg321, Thr324, Gly86, Val248, and Phe264, electrostatic interactions with Glu256, and hydrophobic interactions with His244, Pro284, and Val283. Peptide IIPFIF forms three conventional hydrogen bonds with Asn81, His244, and Asn260 of TYR, van der Waals forces with Pro284, His259, Met257, Cys83, Thr324, and Glu322, hydrophobic interactions with Pro277, Phe264, and Val248, and π-σ bond interactions with Val283. Peptide ILFTLL established four conventional hydrogen bonds with Asn81, Asn260, His244, and Gly281, van der Waals forces with Cu401, Thr324, Tyr65, Ser282, His85, Gly86, Arg268, Met280, and Phe292, π-σ bond interactions with His263, hydrophobic interactions with His61, His259, Pro284, Val248, Phe264, and Ala286, π-π stacking interactions with Glu322. By contrast, the three peptides in this study interacted with the binding sites for His61, His85, His259, His263, and Phe292, reducing their rotational freedom and side chain conformation, interfering with the integrity of their binding sites, and further functioning as TYR inhibitors.

Interestingly, despite only having one key amino acid in its interaction with TYR, the TYR inhibition rate of peptide IIPFIF was the highest of the three peptides, which was consistent with its highest Peptide Ranker score. The TYR inhibition rate of peptide ILFTLL was the lowest of the three peptides, which was consistent with the lowest binding energy of the docking data, despite possessing five critical amino acids in its interaction with TYR. Although TYR has an exterior hydrophilic environment, its active core produces a hydrophobic cavity inside, and amino acids with hydrophobic side chains are more likely to bind in the hydrophobic pocket near TYR’s active site [52]. Meanwhile, aromatic amino acids possess benzene rings that can be buried in the active site while binding to TYR [53], and their hydrophobic nature aids in the stabilization of the binding conformation, resulting in higher inhibitory efficacy against TYR. The difference is that peptide IIPFIF has an aromatic amino acid at the C-terminal and enters the active pocket from the C-terminal, while peptide TIPPPT and peptide ILFTLL have no aromatic amino acid at either end; peptide IIPFIF is composed entirely of hydrophobic amino acids and has the larger portion entered in the active pocket, while the other two peptides have one to two hydrophilic amino acids and have the smaller portion entered in the active pocket. In addition, the aromatic amino acid end of peptide IIPFIF also forms π-σ bond interactions and hydrogen bonding forces with the Val283 and Asn260 sites of the active pocket, respectively. These may be the main causes for the greatest TYR inhibition rate of IIPFIF.

Overall, the TYR inhibitory peptide isolated from the ERJ contacted residues in the active pocket of the TYR, including conventional hydrogen bonding, van der Waals forces, hydrophobic interactions, and π-σ bonds, etc. Hydrophobic and aromatic amino acids are more favorable for binding to the active pocket of TYR. The active peptides occupy the catalytic pocket of TYR and disrupt the binding site integrity, achieving the goal of inhibiting TYR by these mechanisms.

Computer-aided screening design is a new drug development tool based on computational science that has been gradually developed in response to social demands. Based on known protein receptor architectures and features, computer-assisted analysis can assess protein active sites and ligand interactions and generate quantitative conformational relationship models or pharmacophore models to forecast or develop suitable medications. This technique can not only rapidly filter out compounds with no predicted biological action, but it can also cut the cost of pointless R&D owing to blind studies. Subsequent work in this study will focus on rational design to obtain highly active peptides and investigate the structural interactions between peptides and TYR. Furthermore, the absence of animal trials will be compensated for in future experiments to corroborate the in vivo activity.

## 4. Conclusions

The TYR inhibition rate of ERJ was able to reach 90.8% after optimizing the enzymatic digesting procedure. Dextran gel chromatography was used to isolate the fraction D4 with the highest TYR inhibitory activity. Combined with LC-MS/MS and molecular docking analysis, three novel TYR inhibitory peptides, TIPPPT (IC_50_ = 7.59 mg/mL), IIPFIF (IC_50_ = 6.16 mg/mL), and ILFTLL (IC_50_ = 9.25 mg/mL), were screened from the D4 fraction with remarkable inhibitory activity. The docking results indicated that peptides with hydrophobic side chains and aromatic amino acids at the ends were more likely to bind to the hydrophobic pocket near the TYR active site and form a stable conformation, whereas IIPFIF possessed both of these properties, and thus had the strongest TYR inhibitory activity. In addition to the six His, Phe90, and Phe292 sites, Val283 and Asn260 play essential roles in peptide binding to TYR. Based on this study, ERJ can be used to develop functional food or nutraceuticals for TYR inhibitors.

## Figures and Tables

**Figure 1 foods-12-02240-f001:**
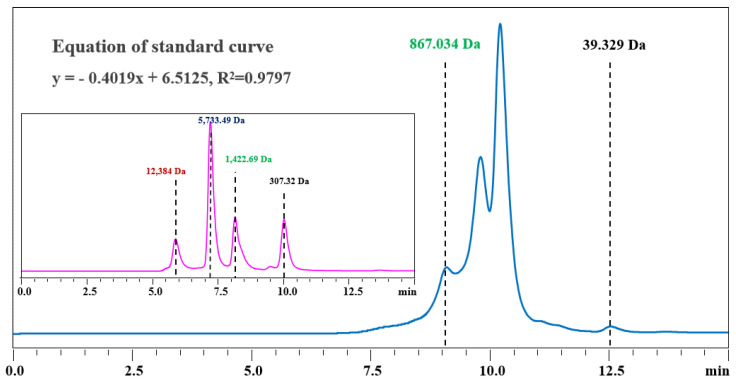
Molecular weight distribution of the standard substance and ERJ.

**Figure 2 foods-12-02240-f002:**
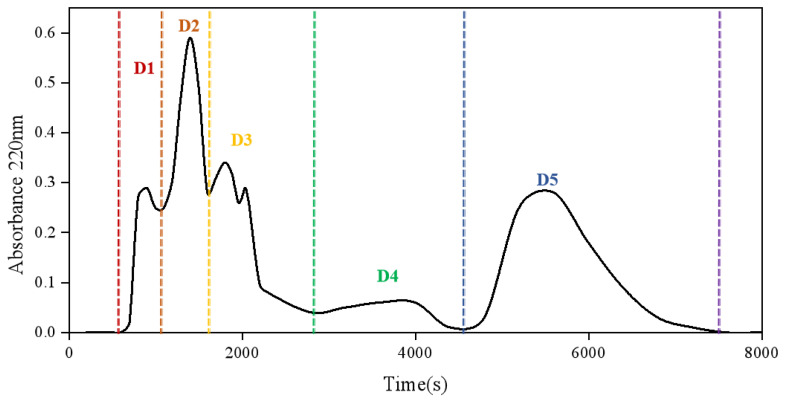
G-15 dextran gel chromatography of ERJ.

**Figure 3 foods-12-02240-f003:**
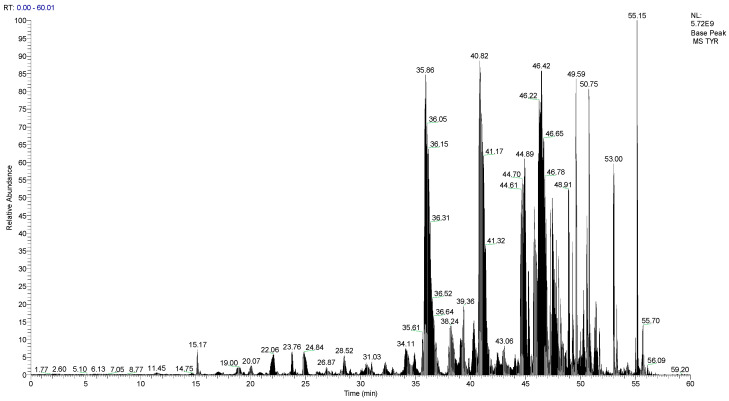
Base peak diagram of F4 component.

**Figure 4 foods-12-02240-f004:**
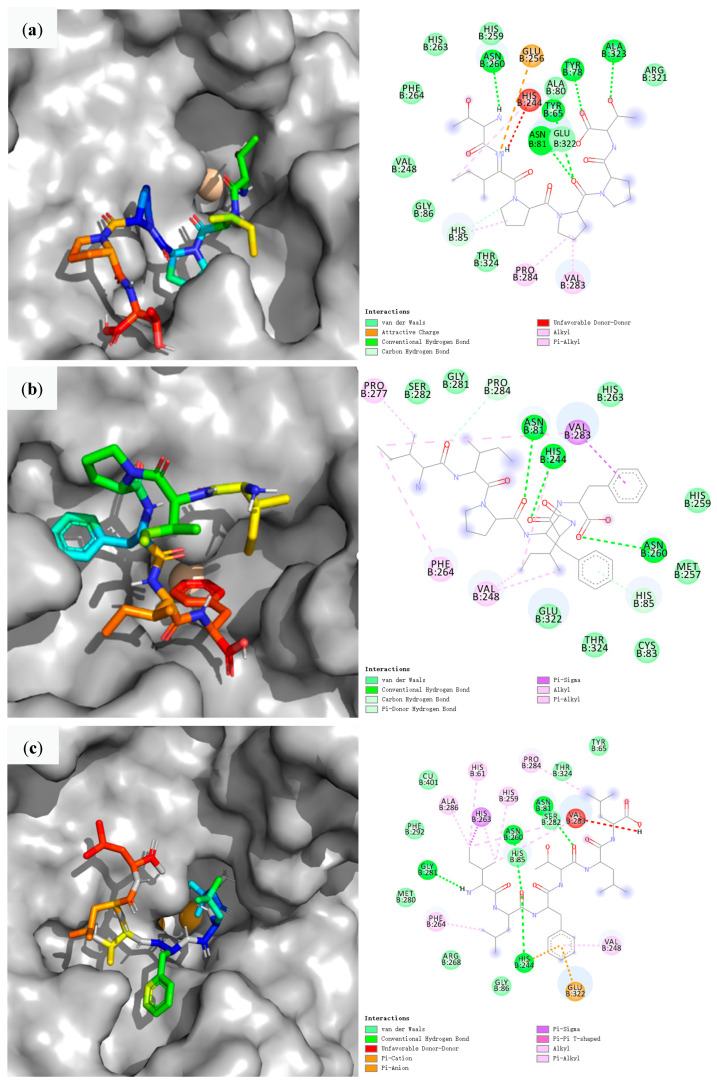
Results of TYR molecular docking with TIPPPT (**a**), IIPFIF (**b**), ILFTLL (**c**).

**Table 1 foods-12-02240-t001:** Find analysis parameter settings.

Item	Value
Enzyme	noEnzyme
Precursor Tolerance (Main search)	4.5 ppm
Precursor Tolerance (First search)	20 ppm
MS/MS Tolerance	20 ppm
Database	UniProt-Apis mellifera (Honeybee) [7460]-40765-20220524
Database pattern	Target-Reverse
PSM FDR	0.01
Protein FDR	0.01

**Table 2 foods-12-02240-t002:** Inhibition rate of different components.

Sample	TYR Inhibition Rate %
D0	7.22 ± 0.45 ^d^
D1	8.28 ± 0.83 ^cd^
D2	9.14 ± 0.70 ^c^
D3	15.31 ± 0.61 ^b^
D4	82.89 ± 0.39 ^a^
D5	5.31 ± 1.15 ^e^

Note: The results are shown as means ± SDs, and significant differences are indicated between different letters of the IC_50_ value results (*p* < 0.05).

**Table 3 foods-12-02240-t003:** Peptide molecular weight and TYR inhibition rate of TIPPPT, IIPFIF, and ILFTLL.

Sequence	IC_50_ (mg/mL)	Molecular Weight
TIPPPT	7.59 ± 0.06 ^b^	735.90
IIPFIF	6.16 ± 0.12 ^c^	852.97
ILFTLL	9.25 ± 0.11 ^a^	845.05

Note: The results are shown as means ± SDs, and significant differences are indicated between different letters of the IC_50_ value results (*p* < 0.05).

## Data Availability

The data presented in this study are available on request from the corresponding author.

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
