# Peer review of "Tyrosinase Inhibitory Peptides from Enzyme Hydrolyzed Royal Jelly: Production, Separation, Identification and Docking Analysis"

_foods, 2023, doi:10.3390/foods12112240_

Round 1
Reviewer 1 Report
This manuscript is dealing with the separation, identification and docking analysis of tyrosinase inhibitory peptides from enzymatic hydrolysis products of royal jelly. This manuscript is very interesting. This manuscript needs some modifications as following:
Title: from enzymatic hydrolysis products of royal jelly or royal jelly protein?
L22: in food products with health-promoting properties
Please use keywords which are not present in the manuscript title.
L27: use AD for the first time presenting in the text.
I think that the different methods using for the preparation of bioactive peptides should be mentioned in the introduction.
What is a neutral protease?
Was the amount of protein measured in the initial royal jelly?
The discussion part of the manuscript needs to be improved. The results are only presented and were not compared with the results of the previous studies.
Table 3: adding the SD of the means will help.
Author Response
Dear Editors and Reviewers,
Thank you for your letter and for the reviewers’ comments concerning our manuscript entitled “Separation, identification and docking analysis of tyrosinase inhibitory peptides from enzymatic hydrolysis products of royal jelly” (ID: foods-2390042). Those comments are all valuable and very helpful for revising and improving our paper, as well as the important guiding significance to our researches. We have studied the comments carefully and have made corrections, which we hope meet with approval. The errors and inappropriate expressions have been modified in the new version. Revised portions are marked in red in the revised paper. The main corrections in the paper and the responds to the reviewer’s comments are as follows:
Reviewer: 1
This manuscript is dealing with the separation, identification and docking analysis of tyrosinase inhibitory peptides from enzymatic hydrolysis products of royal jelly. This manuscript is very interesting. This manuscript needs some modifications as following:
- Title: from enzymatic hydrolysis products of royal jelly or royal jelly protein?
Response: Our study involved the enzymatic digestion of fresh royal jelly. We have supplemented it in the manuscript.
- L22: in food products with health-promoting properties
Response: Thank you very much for your comments. We have revised in the manuscript. These contents could be seen in the revised manuscript page 1 line 28-29.
- Please use keywords which are not present in the manuscript title.
Response: Thank you very much for your advices. We have revised keywords in the manuscript.
Keywords: Royal jelly peptides; Alzheimer's disease; Tyrosinase; LC-MS/MS; Molecular docking
These contents could be seen in the revised manuscript page 1 line 30.
- L27: use AD for the first time presenting in the text.
Response: Thank you very much for your comments. We have revised in the text.
- I think that the different methods using for the preparation of bioactive peptides should be mentioned in the introduction.
Response: Thank you very much for your suggestions. Tyrosinase inhibitory peptide (TIPs) are currently prepared using enzymatic digestion, chemical hydrolysis, microbial fermentation and chemical synthesis, with enzymatic digestion and solid-phase synthesis being the two most commonly used techniques.
These contents could be seen in the revised manuscript page 2 line 70-73.
- What is a neutral protease?
Response: We apologize for the errors in the text. Acid protease was chosen for the experiment in this study. Acid proteases are endopeptidases that preferentially cleave peptide bonds between aromatic or hydrophobic amino acid residues at both ends, especially those formed by aromatic amino acids with other amino acids, resulting in small peptides. This protease have 100 000 unit per grams of enzymatic activity.
These contents could be seen in the revised manuscript page 2 line 85-89.
- Was the amount of protein measured in the initial royal jelly?
Response: Yes, we determined the protein content of fresh royal jelly lyophilized powder by Kjeldahl nitrogen determination, which was about 47.5%.
These contents could be seen in the revised manuscript page 3 line 120.
- The discussion part of the manuscript needs to be improved. The results are only presented and were not compared with the results of the previous studies.
Response: Thank you very much for your comments. The IC50 values of the TYR inhibitory activity of our obtained peptides were compared with those of the reported peptides. The maximum activity was found with the peptides IIPFIF, followed by TIPPPT, and the lowest activity was found with the peptide ILFTLL. The obtained peptide has similar TYR inhibitory action to VLT (IC50 = 5.0 mg/mL).
Reference: Krobthong, S.; Yingchutrakul, Y.; Samutrtai, P.; Choowongkomon, K. The C‐terminally shortened analogs of a hexapeptide derived from Lingzhi hydrolysate with enhanced tyrosinase‐inhibitory activity. Arch Pharm 2021, 354(11), 2100204.
These contents could be seen in the revised manuscript page 8 line 323.
- Table 3: adding the SD of the means will help.
Response: Thank you very much for your advices. We have added the SD values in the modified Table 3.
We appreciate for Editors/Reviewers’ warm work earnestly, and hope that the correction will meet with approval.
Once again, thank you very much for your comments and suggestions.
Yours sincerely,
Jianan Sun
E-mail: sunjianan@ouc.edu.cn

Reviewer 2 Report
Totally speaking, this article is regarding the production, isolation, and purification of the tyrosinase inhibitory peptides originated by the technological process of enzymatic hydrolysis products of the royal jelly. The current study's goal was to analyze the tyrozinase inhibitory peptides and fication of the tyrosinase inhibitory peptides originated by the technological process of enzymatic hydrolysis products of the royal jelly. Analysis of the tyrosinase inhibitory peptides and separation of them into a few peptide fractions by using the size-exclusion chomatography technique was one of the research objectives. The manuscript was suitable for Foods journal, and proposed Special Issue. However, there are some main points that require clarification.
(1) Affiliation section: Please include the matching e-mail addresses of the co-authors listed above, as well as their initials in parenthesis. Please, see the instructions for authors.
(2) Title section and whole manuscript: I advise the writers to change the title in a comprehensible way that differs from what is currently in the literary works database. The sole difference between the supplied title and the literary work's protein source is in the given title (see published paper DOI: 10.1016/j.foodchem.2021.129471). For example, a paper could have the following title: ''Tyrosinase inhibitory peptides from enzyme hydrolyzed royal jelly: production, separation, identification and docking analysis''.
(3) Abstract section
There is a lack of a clearly written research goal, the reason for this kind of research, as well as the main objectives of the research. The methodology needs to be rewritten, specifically specifying which analysis was used to performed enzymatic hydrolysis of royal jelly, which to determine the tirosinase inhibitory activity. To say something about the proteases used, why it is used, where it is usually used, and what the purpose of its application is. It is necessary to amend and supplement the Abstract section.
(4) Introduction section
Please complete the introduction with specific statistical data on the worldwide utilization and production of whole tirosinase inhibitory peptides, as well as their production, isolation, and purification technological methods. The nutritional composition (especially protein content) of royal jelly must be listed in detail, including the citation of the necessary literature. Next, isolate the procedures for obtaining peptide fractions with improved bioactivities (i.e., a few illustrations of how the authors enhanced the characteristics of peptides using various endo- and egzo-peptodases).
(5) Materials and Methods section
Line 98: The protein concentration of the starting material is missing. Please complete the starting substrate with its full protein specification.
Line 99: Neutral protease... Is this protease available commercially? If so, please list it. What kind of protease does the protease used belong to? How many units of enzymatic activity does this protease have?
Line 108: Replace the term "enzymatic digestion" with the new and more appropriate term "enzymatic hydrolysis".
Lines 108-109: In the course of enzymatic hydrolysis, the process parameters are varied: temperature, solid-liquid ratio and enzyme concentration. Please specify the ranges of the analyzed parameters, and be sure to specify the coded values of the process parameters in question. So, for each parameter, give real experimental values and assign them -1, 0 and 1 coordinates.
Line 112: You list the unit of the enzyme, but it is not clear. Please clarify if this refers to the declaration unit and if this is per gram of enzyme or per gram of protein in the substrate that is hydrolyzed. Also, if this is an activity that you have defined yourself, you must specify the method by which the activity was found and give the definition of the activity again. As a reminder, proteolytic activity is most commonly expressed in Anson units. Check how it's done and fill in the missing information accordingly.
Line 119: ...acid protease... Why do you now list acid protease when it was originally neutral protease? Change what is necessary.
There are no reaction conditions under which the enzymatic hydrolysis reaction was carried out. So, what pH was used in the hydrolysis? Was it optimal for the work of the protease used? How was the degree of protein hydrolysis monitored, using which method? Was the reaction carried out in a batch system with stirring? If so, how was the heating done (through the mantle or by heating the vessel on the heating element) and at what speed of mixing (magnetic mixing or mechanical mixing with propellers)? Write everything necessary!
Line 121-122: After the end of the enzymatic hydrolysis, When was the end of the enzymatic hydrolysis? But was it a limited time of 2-3-4 hours, or was that end determined on the basis of what the fixed value of the hydrolysis degree was? Please clarify and explain. Then, after hydrolysis was complete, how much protein was in the system? Was the sample with the peptides dried, or was it further used as such for purification? Write necessary.
The names of the devices (centrifuge and appropriated rotors; freeze-dryer, etc.) that were used in the whole experimental work and their manufacturers must be mentioned.
(6) Weather is conceivable for the authors to rephrase their conclusion section by drawing a general (concluded) influence on examined propeties and amino acid compositions i.e. molecular weight of isolated novel peptides?
(7) It is advised that the authors recheck the main text during the revision to make this manuscript more readable.
Author Response
Dear Editors and Reviewers,
Thank you for your letter and for the reviewers’ comments concerning our manuscript entitled “Separation, identification and docking analysis of tyrosinase inhibitory peptides from enzymatic hydrolysis products of royal jelly” (ID: foods-2390042). Those comments are all valuable and very helpful for revising and improving our paper, as well as the important guiding significance to our researches. We have studied the comments carefully and have made corrections, which we hope meet with approval. The errors and inappropriate expressions have been modified in the new version. Revised portions are marked in red in the revised paper. The main corrections in the paper and the responds to the reviewer’s comments are as follows:
Reviewer: 2
Totally speaking, this article is regarding the production, isolation, and purification of the tyrosinase inhibitory peptides originated by the technological process of enzymatic hydrolysis products of the royal jelly. The current study's goal was to analyze the tyrozinase inhibitory peptides and fication of the tyrosinase inhibitory peptides originated by the technological process of enzymatic hydrolysis products of the royal jelly. Analysis of the tyrosinase inhibitory peptides and separation of them into a few peptide fractions by using the size-exclusion chomatography technique was one of the research objectives. The manuscript was suitable for Foods journal, and proposed Special Issue. However, there are some main points that require clarification.
- Affiliation section
Please include the matching e-mail addresses of the co-authors listed above, as well as their initials in parenthesis. Please, see the instructions for authors.
Response: Thanks a lot for the advice, we have added e-mail addresses and initials of the co-authors in the affiliation section. These contents could be seen in the revised manuscript page 1 line 6-11.
- Title section and whole manuscript
I advise the writers to change the title in a comprehensible way that differs from what is currently in the literary works database. The sole difference between the supplied title and the literary work's protein source is in the given title (see published paper DOI: 10.1016/j.foodchem.2021.129471). For example, a paper could have the following title: ''Tyrosinase inhibitory peptides from enzyme hydrolyzed royal jelly: production, separation, identification and docking analysis''.
Response: Thank you very much for your advice. I have changed the title of this article to “Tyrosinase inhibitory peptides from enzyme hydrolyzed royal jelly: production, separation, identification and docking analysis”.
- Abstract section
There is a lack of a clearly written research goal, the reason for this kind of research, as well as the main objectives of the research. The methodology needs to be rewritten, specifically specifying which analysis was used to performed enzymatic hydrolysis of royal jelly, which to determine the tirosinase inhibitory activity. To say something about the proteases used, why it is used, where it is usually used, and what the purpose of its application is. It is necessary to amend and supplement the Abstract section.
Response: Thank you very much for your valuable comments. I have made changes and additions to the abstract section.
Tyrosinase is inextricably related to the development of Alzheimer's disease. The effects of natural tyrosinase inhibitors on human health have attracted widespread attention. This study aimed to isolate and analyze the tyrosinase (TYR) inhibitory peptides in the enzymatic hydrolysis products of royal jelly. We first analyzed optimal process conditions for the enzymatic digestion of royal jelly by single-factor and orthogonal experiments, and then used gel filtration chromatography to obtain five fractions D1~D5 with molecular weights ranging from 600 to 1100 Da. LC-MS/MS was applied to identify the fractions with the highest activity, and the obtaining peptides were screened and molecularly docked using Autodock Vina. The results showed that the optimal enzymatic conditions for tyrosinase inhibition rate were: acid protease, enzyme addition 10 000 U/g, initial pH 4, feed to liquid ratio 1:4, enzymatic temperature 55 ℃ and enzymatic time 4 h. The D4 fraction had the most significant TYR inhibitory activity. The IC50 values of the three new peptides with the strongest TYR inhibitory activity, TIPPPT, IIPFIF and ILFTLL, were obtained as 7.59 mg/mL, 6.16 mg/mL and 9.25 mg/mL, respectively. The molecular docking results showed that aromatic and hydrophobic amino acids were more favorable to occupy the catalytic center of TYR. In conclusion, the new peptide extracted from royal jelly has the potential to be used as a natural TYR inhibitory peptide in health foods.
These contents could be seen in the revised manuscript page 1 line 13-29.
- Introduction section
Please complete the introduction with specific statistical data on the worldwide utilization and production of whole tirosinase inhibitory peptides, as well as their production, isolation, and purification technological methods. The nutritional composition (especially protein content) of royal jelly must be listed in detail, including the citation of the necessary literature. Next, isolate the procedures for obtaining peptide fractions with improved bioactivities (i.e., a few illustrations of how the authors enhanced the characteristics of peptides using various endo- and egzo-peptodases).
Response: Thank you very much for your comments.
Tyrosinase inhibitory peptides (TIPs) generated from food are mostly utilized in food, pharmaceutical and skin care applications, particularly for whitening and photodamage restoration, and are appreciated for their excellent biosafety and ease of absorption. Food protein-derived TIPs have strong biocompatibility and are prevalent in mammalian milk, agricultural products and aquatic products , primarily from terrestrial and aquatic sources. TIPs are currently prepared using enzymatic digestion, chemical hydrolysis, microbial fermentation and chemical synthesis, with enzymatic digestion and solid-phase synthesis being the two most commonly used techniques. Separation and purification methods primarily include ultrafiltration (UF), gel filtration chromatography (GFC) [9-10] and reversed-phase high performance liquid chromatography (RP-HPLC).
Fresh royal jelly mainly consists of water (60–70%), protein (9–18%), carbs (7–18%), lipids (3–8%), minerals (1%–3%), vitamins, and a few unidentified substances. Acid protease, papain, alkaline protease, and other proteases are often employed for enzymatic digestion of royal jelly. They are all endopeptidases that can cleave peptide bonds generated by a wide range of amino acids. Acid protease, for illustration, can cleave the peptide link between aromatic or hydrophobic amino acid residues on both ends.
These contents could be seen in the revised manuscript page 2 line 65-75, 79-81 and 85-89.
5、Materials and Methods section
(1) Line 98: The protein concentration of the starting material is missing. Please complete the starting substrate with its full protein specification.
Response: Thanks for your suggestions. We used fresh royal jelly as the substrate for enzymatic digestion and determined the protein content of fresh royal jelly lyophilized powder by Kjeldahl nitrogen determination, which was about 47.5%.
These contents could be seen in the revised manuscript page 3 line 120.
- Line 99: Neutral protease... Is this protease available commercially? If so, please list it. What kind of protease does the protease used belong to? How many units of enzymatic activity does this protease have?
Response: We apologize for the errors in the text and thank you for your comments. Acid protease was chosen for the experiment in this study. Our acid protease is purchased from Pangbo Biological Engineering Co., ltd. (Nanning, China) as a food additive, which can be widely used in food industry (enzymatic digestion of animal and plant proteins produces nutrient solution to enhance the nutritional value of products), alcohol industry (using acid protease in wine fermentation can promote yeast growth, speed up fermentation and increase alcohol yield) and feed industry (enzyme-containing feed can improve the nutritional value of feed and promote the growth and weight gain of livestock and poultry), etc. Acid proteases are endopeptidases that preferentially cleave peptide bonds between aromatic or hydrophobic amino acid residues at both ends, especially those formed by aromatic amino acids with other amino acids, resulting in small peptides. This protease have 100 000 unit per grams of enzymatic activity.
- Line 108: Replace the term "enzymatic digestion" with the new and more appropriate term "enzymatic hydrolysis".
Response: Thank you very much for your comments. We have revised in the manuscript.
- Lines 108-109: In the course of enzymatic hydrolysis, the process parameters are varied: temperature, solid-liquid ratio and enzyme concentration. Please specify the ranges of the analyzed parameters, and be sure to specify the coded values of the process parameters in question. So, for each parameter, give real experimental values and assign them -1, 0 and 1 coordinates.
Response: Thank you very much for your advices. We established a three-factor, three-level Box-Benhnken central combination test with tyrosinase inhibition as the response value, and the levels of each factor were coded using -1, 0, and 1. We have added this section to the Supplementary file.
Table S1. Factors and levels of the Box-Behnken experiment design
|
Factors |
Levels |
||
|
-1 |
0 |
1 |
|
|
Solid-liquid ratio |
1:3 |
1:5 |
1:7 |
|
Temperature (℃) |
30 |
50 |
70 |
|
Enzyme dose (U/g) |
4000 |
7000 |
10000 |
- Line 112: You list the unit of the enzyme, but it is not clear. Please clarify if this refers to the declaration unit and if this is per gram of enzyme or per gram of protein in the substrate that is hydrolyzed. Also, if this is an activity that you have defined yourself, you must specify the method by which the activity was found and give the definition of the activity again. As a reminder, proteolytic activity is most commonly expressed in Anson units. Check how it's done and fill in the missing information accordingly.
Response: Thanks very much for your comments. “U/g” is the international unit of enzyme activity force. A unit of acid protease activity, denoted as U/g, is the amount of casein hydrolyzed in 1 minute at 40°C and pH 3.0 to create 1 microgram of tyrosine from 1 gram of solid enzyme powder.
(6) Line 119: ...acid protease... Why do you now list acid protease when it was originally neutral protease? Change what is necessary.
Response: We apologize for the errors in the text. Acid protease was chosen for the experiment in this study.
- There are no reaction conditions under which the enzymatic hydrolysis reaction was carried out. So, what pH was used in the hydrolysis? Was it optimal for the work of the protease used? How was the degree of protein hydrolysis monitored, using which method? Was the reaction carried out in a batch system with stirring? If so, how was the heating done (through the mantle or by heating the vessel on the heating element) and at what speed of mixing (magnetic mixing or mechanical mixing with propellers)? Write everything necessary!
Response: Thanks very much for your comments. We have completed the reaction conditions for enzymatic digestion as well as the necessary instrumentation information. The single-factor experimental procedure was not demonstrated in this study. The optimal model conditions were obtained as follows: acid protease, material-liquid ratio 1: 4, temperature 55℃, enzyme addition 10 000 U/g, initial pH 4.0 and enzymatic digestion time 4h. The temperature was controlled with a THZ-82 thermostatic water bath shaker (Shanghai Zhi Chu Instruments co. Ltd., Shanghai, China), and the gyration frequency was kept constant at 200rpm/min throughout the enzymatic digestion process. After enzymatic digestion, the ERJ was boiled at 100℃ for 12 min to deactivate the enzyme. Because there was no direct correlation between the degree of hydrolysis and the TYR inhibition rate, we did not use the degree of hydrolysis to evaluate the hydrolysis effect. We used the inhibition rate of TYR as an indicator to determine the degree of enzymatic cleavage of ERJ.
These contents could be seen in the revised manuscript page 3 line 134-140.
- Line 121-122: After the end of the enzymatic hydrolysis, When was the end of the enzymatic hydrolysis? But was it a limited time of 2-3-4 hours, or was that end determined on the basis of what the fixed value of the hydrolysis degree was? Please clarify and explain. Then, after hydrolysis was complete, how much protein was in the system? Was the sample with the peptides dried, or was it further used as such for purification? Write necessary.
Response: yes, Our preliminary experiments explored enzymatic digestion times of 3-7 h, and finally 4 h was chosen as the optimal time. The supernatant was measured after centrifugation of ERJ using TYR inhibition rate as an indicator. We focused preferentially on TYR activity and not on protein hydrolysis. R0 was obtained by freeze-drying the ERJ prior to gel filtration chromatography.
- The names of the devices (centrifuge and appropriated rotors; freeze-dryer, etc.) that were used in the whole experimental work and their manufacturers must be mentioned.
Response: Thanks very much for your comments. We have added to this in the text.
6、Weather is conceivable for the authors to rephrase their conclusion section by drawing a general (concluded) influence on examined propeties and amino acid compositions i.e. molecular weight of isolated novel peptides?
Response: Thanks very much for your comments. Because the three novel peptides in this study are all 6-peptides with rather minor molecular weight changes, they hardly reflect any particular pattern. Therefore, we chose to analyze them in terms of hydrophobic properties and classes of amino acids. We have rephrased the conclusion section.
The TYR inhibition rate of ERJ was able to reach 90.8% after optimizing the enzymatic digesting procedure. Dextran gel chromatography was used to isolate the fraction D4 with the highest TYR inhibitory activity. Combined with LC-MS/MS and molecular docking analysis, three novel TYR inhibitory peptides, TIPPPT (IC50 = 7.59 mg/mL), IIPFIF (IC50 = 6.16 mg/mL ) and ILFTLL (IC50 = 9.25 mg/mL), were screened from the D4 fraction with remarkable inhibitory activity. The docking results indicated that peptides with hydrophobic side chains and aromatic amino acids at the ends were more likely to bind to the hydrophobic pocket near the TYR active site and form a stable conformation, whereas IIPFIF possessed both of these properties and thus had the strongest TYR inhibitory activity. In addition to the six His, Phe90, and Phe292 sites, Val283 and Asn260 play essential roles in peptide binding to TYR. Based on this study, ERJ can be used to develop functional food or nutraceuticals for TYR inhibitors.
These contents could be seen in the revised manuscript page 11 line 420-431.
7、It is advised that the authors recheck the main text during the revision to make this manuscript more readable.
Response: Thank you very much for your comments. We have rechecked the main text and made some necessary changes.
We appreciate for Editors/Reviewers’ warm work earnestly, and hope that the correction will meet with approval.
Once again, thank you very much for your comments and suggestions.
Yours sincerely,
Jianan Sun
E-mail: sunjianan@ouc.edu.cn

Round 2
Reviewer 1 Report
Accept